# SPACETIME $E(n)$-TRANSFORMER: EQUIVARIANT ATTENTION FOR SPATIO-TEMPORAL GRAPHS

## ABSTRACT

We introduce an $E(n)$-equivariant Transformer architecture for spatio-temporal graph data. By imposing rotation, translation, and permutation equivariance inductive biases in both space and time, we show that the Spacetime $E(n)$-Transformer (SET) outperforms purely spatial and temporal models without symmetry-preserving properties. We benchmark SET against said models on the $N$-body problem, a simple physical system with complex dynamics. While existing spatio-temporal graph neural networks focus on sequential modeling, we empirically demonstrate that leveraging underlying domain symmetries yields considerable improvements for modeling dynamical systems on graphs.

## 1 INTRODUCTION

Many problems that we wish to model with neural networks possess underlying geometric structure with symmetries. *Geometric Deep Learning*, a term coined in the seminal work of Bronstein et al. (2021), is an Erlangen program for deep learning that systematizes inductive biases as group symmetries $G$, arising through notions of invariance and equivariance.

Recent work, like $SE(3)$-Transformers Fuchs et al. (2020) and $E(n)$-Graph Neural Networks Satorras et al. (2022), impose different notions of group equivariance on neural networks to inform architecture choice. While these neural network architectures encode spatial inductive biases, they notably lack a time component. Temporal Graph Networks Rossi et al. (2020) proposed an efficient framework that learns from dynamic graphs. However, this architecture assumes the topology of graphs changes over time. In this paper, we discuss *spatio-temporal graphs*, which have a fixed topology with changing features over discrete time steps. Recent works Jin et al. (2023), Marisca et al. (2022), Cini et al. (2023) have treated node features as time series and edges as the relationships between these series. Such spatio-temporal graph neural networks (STGNNs) have a plethora of applications, from simulating biomolecular interactions to modeling financial time series.

Similarly, while STGNNs improve representation learning of sequential graph data, minimal research has been done on preserving group symmetries in a spatio-temporal fashion. In particular, sequential models ought to preserve spatial group symmetries at each time step. Famously, Noether's first theorem formalizes the notion of infinitesimal symmetries of the so-called Lagrangian of a physical system, in terms of perturbations with respect to both space and time, by determining conserved quantities. Inspired by this intuition of temporal and spatial symmetry, we seek to derive a neural network architecture that is equivariant in both temporal and spatial components.

Classical neural network architectures like RNNs are discrete approximations to continuous time-domain signals, obeying a differential equation with respect to time. If an RNN is invariant to *time-warping*, a monotonically increasing and differentiable function of time, it takes the form of an LSTM Bronstein et al. (2021), which unlike a vanilla RNN, captures long-term dependencies. Similarly, the dynamics of classical physical systems satisfy the Euler-Lagrange equations, i.e. the equations of motion. Hence, we use the $N$-body problem, as described in Trenti & Hut (2008) and alluded to in Satorras et al. (2022), as an ideal candidate to test our hypothesis that preserving *G-equivariance ameloriates long-term spatio-temporal graph modeling*. We will use a Transformer for the temporal component of the architecture, preserving long-term dependencies and, hence, invariance to time-warping. Each node of the graph will have features, coordinates, and velocities. As such, the neural network should be equivariant under rotational and translational symmetries $E(n)$

acting on coordinates. It should also be equivariant with respect to rotational symmetries $SO(n)$ acting on velocities. Lastly, the nodes should be permutation equivariant.

## 2 BACKGROUND

### 2.1 GEOMETRIC DEEP LEARNING

Following the insights of Geometric Deep Learning Bronstein et al. (2021), the input signals to machine learning models have an underlying domain $\Omega$. Examples of such domains include grids, graphs, and manifolds. The space of signals over $\Omega$ possesses a vector-space structure. That is Bronstein et al. (2021):

**Definition 2.1.** The space of $\mathcal{C}$-valued signals on $\Omega$ is

$$\mathcal{X}(\Omega, \mathcal{C}) = \{x : \Omega \to \mathcal{C}\},$$

which is a vector space of functions.

The symmetry of the domain $\Omega$ will impose structure on the signal space $\mathcal{X}(\Omega)$, thus inducing structure on the space of interpolants

$$\mathcal{F}(\mathcal{X}(\Omega)) = \{f_{\theta \in \Theta}\}$$

for $f_\theta$ a neural network. In what follows, we canonically refer to $\mathcal{X}(\Omega)$ as $V$ for brevity.

### 2.2 GROUP REPRESENTATIONS, INVARIANCE, AND EQUIVARIANCE

**Definition 2.2.** A *representation* of a group $G$ on a vectorspace $V$ over a field $K$ is a homomorphism

$$\rho : G \to GL(K, V)$$

such that $\rho(gh) = \rho(g)\rho(h)$ for all $g \in G$, $h \in G$, where $GL(K, V)$ is the general linear group of automorphisms $\varphi : V \xrightarrow{\sim} V$, i.e., the set of bijective linear transformations with function composition as its binary operation.

In this paper, we are interested in the group of rotational symmetries $SO(n)$ and the group of isometries $E(n)$ of $\mathbb{R}^n$, as these are the naturally-induced symmetries of particles. Rotations are distance, angle, and orientation preserving transformations. The group of rotations in $n$ dimensions is

$$SO(n) = \{Q \in M_n(\mathbb{R}) | Q^\top Q = I \text{ and } \det Q = +1\},$$

where $M_n(\mathbb{R})$ is the set of $n \times n$ matrices with entries in $\mathbb{R}$. We represent a group element $g \in SO(n)$ with $\rho(g) \in GL(\mathbb{R}, \mathbb{R}^n)$, acting on $\mathbf{x} \in \mathbb{R}^n$ as $\rho(g) : \mathbf{x} \mapsto Q\mathbf{x}$ where $Q \in \mathbb{R}^{n \times n}$ is an orthogonal matrix (see Appendix A for more details). We restrict the notion of equivariance to functions of Euclidean space, as will be the case for neural networks. In Appendix A, we provide a more general definition.

**Definition 2.3.** A function $f : \mathbb{R}^n \to \mathbb{R}^n$ is $SO(n)$-equivariant if

$$Qf(\mathbf{x}) = f(Q\mathbf{x})$$

for all $Q \in \mathbb{R}^{n \times n}$ orthogonal and $\mathbf{x} \in \mathbb{R}^n$.

The Euclidean group $E(n)$ is the set of isometries of Eucliden space $\mathbb{R}^n$, i.e. transformations that preserve distance between points, represented as a rotation followed by a translation. More precisely, $E(n) = \{\varphi : \mathbb{R}^n \to \mathbb{R}^n | \varphi \text{ isometry}\}$. We represent a group element $g \in E(n)$ with $\rho(g) \in GL(\mathbb{R}, \mathbb{R}^n)$, acting on $\mathbf{x} \in \mathbb{R}^n$ as $\rho(g) : \mathbf{x} \mapsto Q\mathbf{x} + \mathbf{b}$ where $Q \in \mathbb{R}^{n \times n}$ is an orthogonal rotation matrix and $\mathbf{b} \in \mathbb{R}^n$ is a translation vector. Again, we provide a definition of equivariance with respect to functions of Euclidean space.

**Definition 2.4.** A function $f : \mathbb{R}^n \to \mathbb{R}^n$ is $E(n)$-equivariant if

$$Qf(\mathbf{x}) + \mathbf{b} = f(Q\mathbf{x} + \mathbf{b})$$

for all $Q \in \mathbb{R}^{n \times n}$ orthogonal rotation matrices, $\mathbf{b} \in \mathbb{R}^n$ translation vectors, and for all $\mathbf{x} \in \mathbb{R}^n$.

## 3 METHOD

In this paper, we are interested in physical systems that can be modelled as a sequence of graphs $G_t = (\mathcal{V}_t, \mathcal{E}_t)$ for $t = 1, \ldots, L$ with nodes $v_i(t) \in \mathcal{V}_t$ and edges $e_{ij}(t) \in \mathcal{E}_t$. In particular, we seek to model the dynamics of the $N$-body problem Trenti & Hut (2008). For this task, we assume a priori that the graph is complete since a charged particle will interact with every other particle in a Van der Waals potential under Coulomb's law. Similarly, a mass will interact with every other charged particle in a gravitational potential under Newton's law of universal gravitation. In addition, we assume that particles are neither created nor destroyed as the system evolves in time, so the nodes $\mathcal{V}_t$ in the graph remain the same. Let $\mathcal{G} := (G_t)_{1 \leq t \leq L}$ be a sequence of topologically-identical graphs with changing features, known as a *spatio-temporal graph*. The task under consideration is learning a function that predicts the associated features of graph. In particular, given $\mathcal{G}$, we are interested in predicting the positions and velocities of all masses in the system after $H$ additional time steps where $H >> L$.

To equip the spatio-temporal model of mass interactions with the appropriate inductive biases, we leverage both spatial and temporal notions of attention. For node $i$ at time step $t$ to attend to all the past neighborhoods of that node, we need to (1) aggregate nodes spatially to obtain spatially-contextual embeddings and (2) obtain temporally-contextual embeddings via temporal aggregation.

We fix a time slice $t$ such that the features derive from $G_t = (\mathcal{V}_t, \mathcal{E}_t)$. From the current features $\mathbf{h}_i^{(l)}(t)$ of node $i$ at layer $l$, we form the next layer features $\mathbf{h}_i^{(l+1)}(t)$ by aggregating neighboring node features. In particular,

$$\mathbf{h}_i^{(l+1)}(t) = \phi\left(\mathbf{h}_i^{(l)}(t), \bigoplus_{j \in \mathcal{N}_i} a(\mathbf{h}_i^{(l)}(t), \mathbf{h}_j^{(l)}(t))\psi(\mathbf{h}_j^{(l)}(t))\right), \tag{1}$$

where $\oplus$ is a permutation-invariant function Bronstein et al. (2021), and $a$ is a self-attention mechanism, often a normalized softmax across neighbors.

### 3.1 $E(n)$-EQUIVARIANT SPATIAL ATTENTION

Satorras et al. (2022) introduced $E(n)$-Equivariant Graph Neural Networks (EGNNs). Every node in the graph $G = (\mathcal{V}, \mathcal{E})$ has features $\mathbf{h}_i \in \mathbb{R}^d$ and coordinates $\mathbf{x}_i \in \mathbb{R}^n$. In addition, we keep track of each mass's velocity $\mathbf{v}_i \in \mathbb{R}^n$. The Equivariant Graph Convolutional Layer (EGCL) takes the set of node embeddings $h^{(l)} = \left\{\mathbf{h}_1^{(l)}, \ldots, \mathbf{h}_N^{(l)}\right\}$, coordinate embeddings $x^{(l)} = \left\{\mathbf{x}_1^{(l)}, \ldots, \mathbf{x}_N^{(l)}\right\}$, velocity embeddings $v^{(l)} = \left\{\mathbf{v}_1^{(l)}, \ldots, \mathbf{v}_N^{(l)}\right\}$, and edge information $\mathcal{E} = (e_{ij})$ as input and produces the embeddings of the next layer. That is, $h^{(l+1)}, x^{(l+1)}, v^{(l+1)} = \text{EGCL}[h^{(l)}, x^{(l)}, v^{(l)}, \mathcal{E}]$, defined as follows Satorras et al. (2022):

$$\mathbf{m}_{ij} = \phi_e\left(\mathbf{h}_i^{(l)}, \mathbf{h}_j^{(l)}, \|\mathbf{x}_i^{(l)} - \mathbf{x}_j^{(l)}\|_2^2, a_{ij}\right)$$

$$\mathbf{v}_i^{(l+1)} = \phi_v(\mathbf{h}_i^{(l)})\mathbf{v}_i^{(l)} + C\sum_{j \neq i}(\mathbf{x}_i^{(l)} - \mathbf{x}_j^{(l)})\phi_x(\mathbf{m}_{ij})$$

$$\mathbf{x}_i^{(l+1)} = \mathbf{x}_i^{(l)} + \mathbf{v}_i^{(l+1)} \tag{2}$$

$$\mathbf{m}_i = \sum_{j \neq i}\mathbf{m}_{ij}$$

$$\mathbf{h}_i^{(l+1)} = \phi_h(\mathbf{h}_i^{(l)}, \mathbf{m}_i)$$

where $a_{ij}$ are the edge attributes, e.g. the edge values $e_{ij}$, and $\phi_e : \mathbb{R}^{2d+2} \to \mathbb{R}^h$, $\phi_v : \mathbb{R}^d \to \mathbb{R}$, $\phi_x : \mathbb{R}^h \to \mathbb{R}$, and $\phi_h : \mathbb{R}^{d+h} \to \mathbb{R}^{d'}$ are MLPs. In what follows, we assume $d' = d$ for clarity.

Satorras et al. (2022) proved that this layer is equivariant to rotations and translations on coordinates and equivariant to rotations on velocities:

$$\mathbf{h}_i^{(l+1)}, Q\mathbf{x}_i^{(l+1)} + \mathbf{b}, Q\mathbf{v}_i^{(l+1)} = \text{EGCL}[\mathbf{h}_i^{(l)}, Q\mathbf{x}_i^{(l)} + \mathbf{b}, Q\mathbf{v}_i^{(l)}, \mathcal{E}] \tag{3}$$

for $Q \in \mathbb{R}^{n \times n}$ an orthogonal rotation matrix and $\mathbf{b} \in \mathbb{R}^n$ a translation vector. The EGCL is also permutation equivariant with respect to nodes $\mathcal{V}$.

The SchNet Schütt et al. (2018) architecture uses continuous-filter convolutional layers to predict chemical properties of molecules and materials. We use such a continuous filter to model the effect of interactions of nodes on features, which is necessary due to the non-uniform topology of graphs in the $N$-body problem. In layer $l$ of SchNet, for node-wise representations $H^l$, the interactions of a particle $i$ is given by the convolution with neighboring particles:

$$\mathbf{h}_i^{(l+1)} := (H^l * W^l) = \sum_{j=1}^{N} \mathbf{h}_j^{(l)} \circ W_\theta^l \begin{bmatrix} e_1(\mathbf{x}_j - \mathbf{x}_i) \\ \vdots \\ e_n(\mathbf{x}_j - \mathbf{x}_i) \end{bmatrix} \tag{4}$$

where $\circ$ is element-wise multiplication and we expand the distances in a Gaussian basis:

$$e_k(\mathbf{x}_j - \mathbf{x}_i) = \exp(-\gamma(||\mathbf{x}_j - \mathbf{x}_i||_2 - \mu_k)^2) \tag{5}$$

with centers $\mu_k$, chosen between 0 and a cutoff radius Schütt et al. (2018). Moreover, $W_\theta^l : \mathbb{R}^n \to \mathbb{R}^d$ is a filter-generating network, which is learned from the positional data, representing the effect of interactions between nodes on features. It is parameterized as an MLP and takes the radial vectors $\mathbf{x}_j - \mathbf{x}_i$, from node $i$ to $j$, as the input. Futhermore, since we use the Gaussian basis expansion, this interaction convolution is $E(n)$-invariant (see Appendix A) and, thus, the features are preserved under the actions of $E(n)$, as desired.

Since we have a sequence of graphs $\mathcal{G} = \{G_t\}_{1 \le t \le L}$, for a time slice $t$, we apply $K_t$ such EGCL and SchNet transformation layers to the graph $G_t \in \mathcal{G}$:

$$\mathbf{h}_i^{(l+1)}(t) = \text{SchNet}[\mathbf{h}_i^{(l+1)}(t), \mathbf{x}_i^{(l+1)}(t), \mathcal{E}(t)]$$
$$\mathbf{h}_i^{(l+1)}(t), \mathbf{x}_i^{(l+1)}(t), \mathbf{v}_i^{(l+1)}(t) = \text{EGCL}[\mathbf{h}_i^{(l)}(t), \mathbf{x}_i^{(l)}(t), \mathbf{v}_i^{(l)}(t), \mathcal{E}(t)], \tag{6}$$

for $l = 1, \ldots, K_t$. Thus, we obtain *spatially-contextual* representations for node $i$ at time $t$ defined as $\boldsymbol{\theta}_i(t) = \mathbf{h}_i^{(K_t)}(t) \in \mathbb{R}^d$, $\boldsymbol{\xi}_i(t) = \mathbf{x}_i^{(K_t)}(t) \in \mathbb{R}^n$, $\boldsymbol{\omega}_i(t) = \mathbf{v}_i^{(K_t)}(t) \in \mathbb{R}^n$ for $t = 1, \ldots, L$.

## 3.2 TEMPORAL ATTENTION FOR GRAPHS

The objective of this section is to obtain strong *temporally-contextual* representations of the spatial graph embeddings. In the $N$-body problem, we are essentially solving the forward-time Euler-Lagrange equations, a second-order partial differential equation. However, for a fixed node on the spatio-temporal graph, the feature, position, and velocity form a time-series, for which RNN's capture short-term dependencies. It was shown in Tallec & Ollivier (2018) that while vanilla RNNs are not time-warping invariant, LSTMs are a class of such time-warping invariant functions modeling a continuous time-domain signal. Employing this philosophy, the use of an attention-based Transformer architecture to model spatio-temporal graph data merits investigation.

## 3.3 $E(n)$-EQUIVARIANT ATTENTION-BASED TEMPORAL MESSAGE PASSING

We would like the temporal attention to retain the equivariant properties described in Section 3.1. Namely, the Equivariant Temporal Attention Layer (ETAL) should be equivariant to the actions of $E(n)$ on coordinates and the actions of $SO(n)$ on velocities. It should also be permutation equivariant with respect to the actions of the symmetric group $\Sigma_N$ on nodes.

As the EGNN produces feature representations $\mathbf{h}_i(t)$ that are $E(n)$-invariant, we can apply key-query-value self-attention and still preserve the $E(n)$-invariance of features as follows. Define the node-wise query $\mathbf{q}_i(t) = Q_i \boldsymbol{\theta}_i(t)$, key $\mathbf{k}_i(t) = K_i \boldsymbol{\theta}_i(t)$, and value $\mathbf{v}_i(t) = V_i \boldsymbol{\theta}_i(t)$ for $Q_i, K_i, V_i \in \mathbb{R}^{d \times d}$. To reduce memory usage, we share $Q, K, V$ for all nodes. Then the temporally-contextual representation is:

$$\tilde{\boldsymbol{\theta}}_i(t) := \sum_{s=1}^{L} \alpha_i(t, s) \mathbf{v}_i(s) \tag{7}$$

where

$$\alpha_i(t,s) = \frac{\exp(\mathbf{q}_i(t)^\top \mathbf{k}_i(s))}{\sum_{s'=1}^{L} \exp(\mathbf{q}_i(t)^\top \mathbf{k}_i(s'))} \tag{8}$$

Satorras et al. Satorras et al. (2022) showed that for a collection of points $\{\boldsymbol{\xi_i}\}_{i=1}^{N} \in \mathbb{R}^n$, the norm is a unique geometric identifier, such that collections separated by actions of $E(n)$ form an equivalence class. With this in mind, since we desire the attention mechanism for coordinates $\boldsymbol{\xi}_i(t)$ to be equivariant with respect to $E(n)$, we can define the following layer:

$$\mathbf{m}_i(t,s) = \psi_e\left(\boldsymbol{\theta}_i(t), \boldsymbol{\theta}_i(s), ||\boldsymbol{\xi}_i(t) - \boldsymbol{\xi}_i(s)||_2^2\right)$$

$$\tilde{\boldsymbol{\xi}}_i(t) = \boldsymbol{\xi}_i(t) + \sum_{s \in \mathcal{N}(t)\setminus\{t\}} (\boldsymbol{\xi}_i(t) - \boldsymbol{\xi}_i(s))e(t,s) \tag{9}$$

$$= \boldsymbol{\xi}_i(t) + \sum_{\substack{s=1 \\ s \neq t}}^{L} (\boldsymbol{\xi}_i(t) - \boldsymbol{\xi}_i(s))\phi_{\text{inf}}(\mathbf{m}_i(t,s))$$

where $\mathcal{N}(t)$ is the *temporal neighborhood* of time $t$ and, thus, $\mathbf{m}_i(t,s)$ is the message passed from time $s$ to $t$ for node $i$. We use an MLP to parameterize $\Psi_e : \mathbb{R}^n \times \mathbb{R}^n \to \mathbb{R}^h$. Since there is no explicit temporal adjacency matrix, we assume a fully connected temporal graph where a node $i$ at time $t$ exchanges messages with every other time $s$. However, such a fully connected network does not scale and, instead, we infer the edges of the temporal edges in our model. Hence, we use the edge inference of Satorras et al. (2022), whereby $\phi_{\text{inf}} : \mathbb{R}^h \to [0,1]$ takes the edge embedding and provides a soft estimation of its edge value $e(t,s)$.

This is a generalized version of the neighborhood attention described in the $SE(3)$-Transformer network Fuchs et al. (2020) and Tensor Field Network layer Thomas et al. (2018), the intensity function in Zhang et al. (2021), and the invariant point attention in Jumper et al. (2021).

We define an $SO(n)$-equivariant attention layer for velocities $\boldsymbol{\omega}_i(t)$:

$$\tilde{\boldsymbol{\omega}}_i(t) := \sum_{s=1}^{L} \beta_i(t,s)\boldsymbol{\omega}_i(s) \tag{10}$$

where the weight is

$$\beta_i(t,s) = \frac{\boldsymbol{\omega}_i(t)^\top \boldsymbol{\omega}_i(s)}{\sum_{s'=1}^{L} \exp(\boldsymbol{\omega}_i(t)^\top \boldsymbol{\omega}_i(s'))}. \tag{11}$$

In appendix B, we show that the position attention function is $E(n)$-equivariant and the velocity attention function is $SO(n)$-equivariant.

Following the insights of Jin et al. (2023), edges are relationships between time series and they should evolve. Hence, while the adjacency matrix $A \in \mathbb{R}^{N \times N}$ is constant in space when applying ECGL, it should intuitively evolve in time when applying ETAL. That is, if we consider edges as representing the interaction between particles, e.g. the strength of the force, then this must necessarily evolve in time for a non-stationary point cloud system.

We define a key matrix $K(t) = KA(t) \in \mathbb{R}^{N \times N}$, a query matrix $Q(t) = QA(t) \in \mathbb{R}^{N \times N}$, and value matrix $V(t) = VA(t) \in \mathbb{R}^{N \times N}$ for $t = 1, \ldots, L$ and $K, Q, V \in \mathbb{R}^{N \times N}$. Thus, to obtain a temporally-contextual representation of the adjacency matrix at time $t$, we apply attention:

$$\tilde{A}(t) = \sum_{s=1}^{L} \pi(t,s)V(s) \in \mathbb{R}^{N \times N} \tag{12}$$

where

$$\pi(t,s) = \exp(Q(t)^\top K(s))\left(\sum_{s'=1}^{L} \exp(Q(t)^\top K(s'))\right)^{-1}. \tag{13}$$

---

**Algorithm 1** Spatiotemporal Attention (SpatiotempAttn)

---

**Require:** $h, x, v, A$      $\triangleright h \in \mathbb{R}^{L \times N \times d}, x, v \in \mathbb{R}^{L \times N \times n}, A \in \mathbb{R}^{L \times N \times N}$

**Require:** $E : \mathbb{R}^{N \times N} \times \mathbb{R}^{N \times n} \to \mathbb{R}^{N \times (N-1) \times 2}$

**Require:** $W^{[1:L]}, X^{[1:L]}, Y^{[1:L]}, Z^{[1:L]}$    $\triangleright W^{[1:L]} \in \mathbb{R}^{L \times N \times d}, X^{[1:L]} \in \mathbb{R}^{L \times N \times n},$
   $Y^{[1:L]} \in \mathbb{R}^{L \times N \times n}, Z^{[1:L]} \in \mathbb{R}^{L \times N \times N}$

  Initialize MLPs $f_\theta : \mathbb{R}^{L \times N \times d} \to \mathbb{R}^{L \times N \times d}, f_A : \mathbb{R}^{L \times N \times N} \to \mathbb{R}^{L \times N \times N}$

  **for** $t = 1, \ldots, L$ **do**          $\triangleright$ Equivariant Spatial Attention Layer

    $h^{(1)}(t) \leftarrow h(t)$          $\triangleright h^{(1)}(t) \in \mathbb{R}^{N \times d}$

    $x^{(1)}(t) \leftarrow x(t)$          $\triangleright x^{(1)}(t) \in \mathbb{R}^{N \times n}$

    $v^{(1)}(t) \leftarrow v(t)$          $\triangleright v^{(1)}(t) \in \mathbb{R}^{N \times n}$

    $\mathcal{E}(t) \leftarrow E(A(t), x^{(1)}(t))$       $\triangleright \mathcal{E}(t) \in \mathbb{R}^{N \times (N-1) \times 2}$

    **for** $\ell = 1, \ldots, K - 1$ **do**

     $h^{(\ell+1)}(t) = \text{SchNet}[h^{(\ell+1)}(t), x^{(\ell+1)}(t), \mathcal{E}(t)]$

     $h^{(\ell+1)}(t), x^{(\ell+1)}(t), v^{(\ell+1)}(t) = \text{EGCL}[h^{(\ell)}(t), x^{(\ell)}(t), v^{(\ell)}(t), \mathcal{E}(t)]$

    **end for**

    $\theta(t) \leftarrow h^{(K)}(t)$          $\triangleright \theta(t) \in \mathbb{R}^{N \times d}$

    $\xi(t) \leftarrow x^{(K)}(t)$          $\triangleright \xi(t) \in \mathbb{R}^{N \times n}$

    $\omega(t) \leftarrow v^{(K)}(t)$          $\triangleright \omega(t) \in \mathbb{R}^{N \times n}$

  **end for**            $\triangleright$ Equivariant Temporal Attention Layer

    $\theta^{[1:L]} \leftarrow (\theta(1), \ldots, \theta(L))$      $\triangleright \theta^{[1:L]} \in \mathbb{R}^{L \times N \times d}$

    $\xi^{[1:L]} \leftarrow (\xi(1), \ldots, \xi(L))$      $\triangleright \xi^{[1:L]} \in \mathbb{R}^{L \times N \times n}$

    $\omega^{[1:L]} \leftarrow (\omega(1), \ldots, \omega(L))$      $\triangleright \omega^{[1:L]} \in \mathbb{R}^{L \times N \times n}$

    $A^{[1:L]} \leftarrow A$          $\triangleright A^{[1:L]} \in \mathbb{R}^{L \times N \times N}$

    $\hat{\theta}^{[1:L]}, \tilde{\xi}^{[1:L]}, \tilde{\omega}^{[1:L]}, \hat{A}^{[1:L]} = \text{ETAL}\left[\theta^{[1:L]} + W^{[1:L]}, \xi^{[1:L]} + X^{[1:L]}, \omega^{[1:L]} + Y^{[1:L]}, A^{[1:L]} + Z^{[1:L]}\right]$

    $\tilde{\theta}^{[1:L]} = f_\theta(\text{LN}(\hat{\theta}^{[1:L]})) + \hat{\theta}^{[1:L]}$

    $\tilde{A}^{[1:L]} = f_A(\text{LN}(\hat{A}^{[1:L]})) + \hat{A}^{[1:L]}$

    **return** $\tilde{\theta}^{[1:L]}, \tilde{\xi}^{[1:L]}, \tilde{\omega}^{[1:L]}, \tilde{A}^{[1:L]}$

---

Furthermore, in Appendix C, we tensorize the feature, position, velocity, and adjacency components of ETAL to efficiently compute these operations in both space $i = 1, \ldots, N$ and time $t = 1, \ldots, L$ dimensions.

### 3.4 SPACETIME $E(n)$-EQUIVARIANT GRAPH TRANSFORMER

The full spatio-temporal attention module is presented in Algorithm 1. It takes as input the node features $h \in \mathbb{R}^{L \times N \times d}$, positions $x \in \mathbb{R}^{L \times N \times n}$, velocities $v \in \mathbb{R}^{L \times N \times n}$ and adjacency matrices $A \in \mathbb{R}^{L \times N \times N}$. For a spatio-temporal graph $\mathcal{G} = (G_t)_{1 \leq t \leq L}$, we apply an equivariant spatial attention layer in the form of EGCL to obtain spatially-contextual representations $\theta(t) \in \mathbb{R}^{N \times d}, \xi(t) \in \mathbb{R}^{N \times n}, \omega(t) \in \mathbb{R}^{N \times n}$ for $t = 1, \ldots, L$. We share the same EGCL layer across all time steps $t = 1, \ldots, L$. That is, we only learn one set of MLPs $\phi_e, \phi_v, \phi_x$, and $\phi_h$ for each layer across time, which is significantly more memory and parameter efficient.

Observe, at each time step, we apply a transformation $E : \mathbb{R}^{N \times N} \times \mathbb{R}^{N \times n} \to \mathbb{R}^{N \times (N-1) \times 2}$ to the adjacency matrix $A(t) \in \mathbb{R}^{N \times N}$ and the coordinates $x(t) \in \mathbb{R}^{N \times n}$ for $G_t$. This will produce edge attributes $e_{ij}(t) = \left(p_i p_j, \|\mathbf{x}_i(t) - \mathbf{x}_j(t)\|_2^2\right)$ that contain information about particle properties $p$, such as charge or mass, and distance information for neighboring nodes. Since each graph is complete, there are $N \times (N-1)$ such edge attributes, which we store in the tensor $\mathcal{E}(t) \in \mathbb{R}^{N \times (N-1) \times 2}$.

Then we apply equivariant temporal attention in the form of ETAL to the spatial representations $\theta^{[1:L]} \in \mathbb{R}^{L \times N \times d}, \xi^{[1:L]} \in \mathbb{R}^{L \times N \times n}$, and $\omega^{[1:L]} \in \mathbb{R}^{L \times N \times n}$. Feed-forward networks $f_\theta : \mathbb{R}^{L \times N \times d} \to \mathbb{R}^{L \times N \times d}, f_A : \mathbb{R}^{L \times N \times N} \to \mathbb{R}^{L \times N \times N}$ with layer pre-normalization, defined in Appendix D, and residual connection are also applied to the respective feature and edge com-

---

**Algorithm 2** Spacetime $E(n)$-Transformer (SET)

---

**Require:** $h, x, v, A$        $\triangleright\, h \in \mathbb{R}^{L \times N \times d},\, x, v \in \mathbb{R}^{L \times N \times n},\, A \in \mathbb{R}^{N \times N}$

$\hat{\theta}_{(0)}^{[1:L]} \leftarrow h$

$\hat{\xi}_{(0)}^{[1:L]} \leftarrow x$

$\hat{\omega}_{(0)}^{[1:L]} \leftarrow v$

$\hat{A}_{(0)}^{[1:L]} \leftarrow (A, \ldots, A)$        $\triangleright\, \hat{A}_{(0)}^{[1:L]} \in \mathbb{R}^{L \times N \times N}$

**for** $m = 1, \ldots, M$ **do**

     $\hat{\theta}_{(m+1)}^{[1:L]}, \hat{\xi}_{(m+1)}^{[1:L]}, \hat{\omega}_{(m+1)}^{[1:L]}, \hat{A}_{(m+1)}^{[1:L]} \leftarrow \text{SpatiotempAttn}\left(\hat{\theta}_{(m)}^{[1:L]}, \hat{\xi}_{(m)}^{[1:L]}, \hat{\omega}_{(m)}^{[1:L]}, \hat{A}_{(m)}^{[1:L]}\right)$

**end for**

$\hat{x}(L + H) = \frac{1}{L} \sum_{t=1}^{L} \hat{\xi}_{(M)}(t)$        $\triangleright\, \hat{x}(L + H) \in \mathbb{R}^{N \times n}$

$\hat{v}(L + H) = \frac{1}{L} \sum_{t=1}^{L} \hat{\omega}_{(M)}(t)$        $\triangleright\, \hat{x}(L + H) \in \mathbb{R}^{N \times n}$

**return** $\hat{x}(L + H), \hat{v}(L + H)$

---

ponents of the graph. The sinusoidal positional encodings $W^{[1:L]} \in \mathbb{R}^{L \times N \times d}$, $X^{[1:L]} \in \mathbb{R}^{L \times n \times n}$, $Y^{[1:L]} \in \mathbb{R}^{L \times N \times n}$, $Z^{[1:L]} \in \mathbb{R}^{L \times N \times N}$ for the features, positions, velocities, and adjacency matrices are defined in Appendix D.

As the design of spatio-temporal attention is modular, we can continue stacking this architecture as we see fit (see Figure 1). In Algorithm 2, we apply spatio-temporal attention $M$ times. Then we take a mean of the resulting *spatio-temporally contextual* representations of positions and velocities across the time dimension, which we use as the predicted masses' coordinates $\hat{x}(L + H) \in \mathbb{R}^{N \times n}$ and velocities $\hat{v}(L + H) \in \mathbb{R}^{N \times n}$ at the horizon target $t = L + H$.

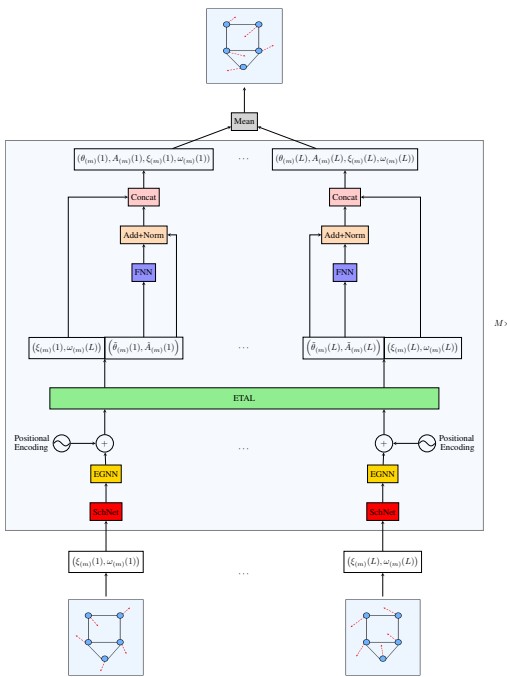

Figure 1: Spacetime $E(n)$-Transformer architecture.

The task is to predict both the positions and velocities of masses at time $L + H$, so we minimize the following loss:

$$\mathcal{L} = ||\hat{x}(L+H) - x(L+H)||_2^2 + \alpha||\hat{v}(L+H) - v(L+H)||_2^2$$

$$= \frac{1}{Nn} \sum_{i=1}^{N} \sum_{j=1}^{n} (\hat{x}_{ij}(L+H) - x_{ij}(L+H))^2 \tag{14}$$

$$+ \frac{\alpha}{Nn} \sum_{i=1}^{N} \sum_{j=1}^{n} (\hat{v}_{ij}(L+H) - v_{ij}(L+H))^2,$$

for $\alpha \in (0,1)$ a hyper-parameter.

## 4 RELATED WORK

Temporal graph learning has a plethora of real-world applications, like COVID-19 contact tracing Chang et al. (2021) Holme (2016) Ding et al. (2021) and misinformation detection Choi et al. (2021) Song et al. (2021) Zhang et al. (2021).

Learning on continuous-time dynamic graphs was introduced by Rossi et al. (2020), which proposed Temporal Graph Networks (TGNs) with a memory module, acting as a summary of what the model has seen so far for each node. Causal Anonymous Walks Wang et al. (2022) is another branch of temporal graph learning, which extracts random walks between edges; however, this is not our focus. Other work like Jin et al. (2023) and Cini et al. (2023) treat node features as time series and edges as correlations between the series. Under this framework, message passing must be able to handle *sequences* of data from the neighborhood of each node, with RNNs Seo et al. (2016), attention mechanisms Marisca et al. (2022), and convolutions Wu et al. (2019).

The Dynamic Graph Convolutional Network (DynGCN) Choi et al. (2021) and DyGFormer Yu et al. (2023) are similar to our method. DynGCN processes each of the graph snapshots with a graph convolutional network to obtain *structural information* and then applies an attention mechanism to capture *temporal information*. Similarly, DyGFormer Yu et al. (2023) learns from historical first-hop neighborhood interactions and applies a Transformer architecture to historical correlations between nodes. However, unlike our paper, DynGCN and DyGFormer do not take into account the inductive biases of the underlying modeling task. Recent work of Wu et al. (2024) presents one of the first spatio-temporally equivariant architectures for graphs, arguing that such an inductive bias allows for the capturing of non-Markovian dynamics of physical systems. Unlike our method, they do not use a continuous convolution for feature extraction. Moreover, they do not consider velocity and assume a static adjacency matrix on which they extract information in the frequency domain by performing a discrete Fourier Transform.

$E(n)$-Equivariant Graph Neural Networks (EGNN) Satorras et al. (2022) defines a model equivariant to the Euclidean group $E(n)$ and, unlike previous methods, does not rely on spherical harmonics such as the $SE(3)$-Transformer Fuchs et al. (2020) and Tensor Field Networks Thomas et al. (2018). The $SE(3)$-Transformer paper briefly alludes to incorporating equivariant attention with an LSTM for temporal causality; however, this is not the primary focus of their work. LieConv Finzi et al. (2020) proposes a framework that allows one to construct a convolutional layer that is equivariant with respect to transformations of a Lie group, equipped with an exponential map. However, the EGNN is simpler and more applicable to problems with point clouds like the $N$-body problem Trenti & Hut (2008) we consider.

As we concern ourselves with modeling a dynamical system, the works of Lagrangian Neural Networks (LNNs) Cranmer et al. (2020) and Hamiltonian Neural Networks (HNNs) Greydanus et al. (2019) are pertinent. HNNs parameterize the Hamiltonian of a system, but require canonical coordinates, which makes it inapplicable to systems where such coordinates cannot be deduced. LLNs parameterize arbitrary Lagrangians of dynamical systems with neural networks, from which it is possible to solve the forward dynamics of the system; however, this requires an additional step of integration, which is cumbersome.

## 5 EXPERIMENTS & RESULTS

### 5.1 DATASET: CHARGED PARTICLES

Adapting the charged $N$-body system dataset from Satorras et al. (2022), we sample $16k$ trajectories for training, $2k$ trajectories for validation, and $2k$ trajectories for testing. Each trajectory has a horizon length of $H = 10k$ and a sequence length of $L = 10$, sampled 10 apart. The task is to predict the positions of particles at time $L + H$. The point cloud consists of $N = 5$ particles, where at each time step, positions $(\mathbf{x}_1(t), \ldots \mathbf{x}_5(t))^\top \in \mathbb{R}^{5 \times 3}$, velocities $(\mathbf{v}_1(t), \ldots \mathbf{v}_5(t))^\top \in \mathbb{R}^{5 \times 3}$, as well as charges $c_1, \ldots, c_5 \in \{-1, +1\}$ are known. The edges between charged particles is $e_{ij}(t) = \left(c_i c_j, ||\mathbf{x}_i(t) - \mathbf{x}_j(t)||_2^2\right)$. We input these known values into SET with features chosen as $h_i(t) = ||\mathbf{v}_i(t)||_2$ for $i = 1, \ldots, 5$. We conducted a hyper-parameter optimization with 30 trials and selected the best model settings, as per Appendix E.

### 5.2 ABLATION STUDY: EQUIVARIANCE, ADJACENCY, AND ATTENTION

Furthermore, we conduct an ablation study on SET, shown in Table 1, which compares the use of equivariance, temporal attention for the adjacency matrix as per Section 3.3, spatial attention, and temporal attention. By selecting the best model on the validation set, we find that incorporating equivariance, spatial attention, and temporal attention enhances performance, whereas using adjacency diminishes it. We hypothesize that the insignificance of temporal adjacency is due to the fact the edge attribute contains information about charges, which does not evolve in time, and information about the distance between particles, which already implicitly exists in the coordinate information.

| Ablation | Model | Params | Val MSE | Test MSE | MSE Ratio |
|----------|-------|--------|---------|----------|-----------|
| Equivariance | **Equiv=True**, Adj=False, SATT=True, TATT=True | 796,058 | **1.21e-10** | **1.25e-10** | − |
| | **Equiv=False**, Adj=False, SATT=True, TATT=True | 796,244 | 1.96e-10 | 2.03e-10 | 1.57× |
| Adjacency | Equiv=True, **Adj=True**, SATT=True, TATT=True | 810,458 | 1.12e-09 | 1.29e-10 | 8.96× |
| Attention | Equiv=True, Adj=False, **SATT=True**, **TATT=False** | 796,049 | 2.73e-10 | 3.57e-10 | 2.86× |

Table 1: Ablation study of equivariance, adjacency, and attention for $N = 20$. We present the model settings, parameter counts, validation & test MSE, and the MSE ratio, which is the ratio of the ablation model's test MSE divided by the best performing model's test MSE.

### 5.3 BASELINES & SCALING $N$

We compare our best performing SET model with optimized LSTM, EGNN, MLP and Linear baselines (see Appendix E for implementation details). SET outperforms all baselines for $N = 5$, as seen in Table 2.

| Model | Params | Test MSE |
|-------|--------|----------|
| SET | 796,058 | **1.25e-10** |
| LSTM | 826,313 | 2.03e-08 |
| EGNN | 100,612 | 2.05e-06 |
| MLP | 67,718 | 3.48e-06 |
| Linear | 3 | 3.04 |

Table 2: Baselines for $N = 5$.

Since the $N = 5$ system is seemingly too simple a task, we scale the dataset to $N = 20$ and $N = 30$. As shown in Figure 2, test MSE remains consistent for all models regardless of the number of charged particles $N$, which is a desirable property. Per Figure 2, the number of model parameters remains

constant for the EGNN, MLP and Linear baselines. Fortunately, the number of parameters in SET also remains constant for all $N$; this is an artifact of the attention layers only being functions of the feature and coordinate dimensions. However, the adjacency attention layer is a function of $N$, which is turned off. Note, the number of parameters in the LSTM increases from $8.2e5$ to $1.8e6$, which is an undesirable property. Further results are included in Appendix E.

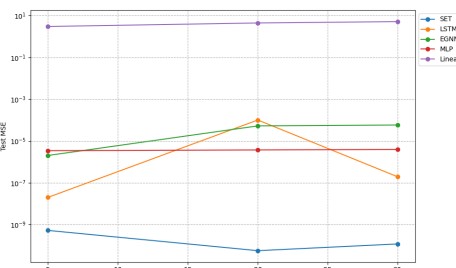 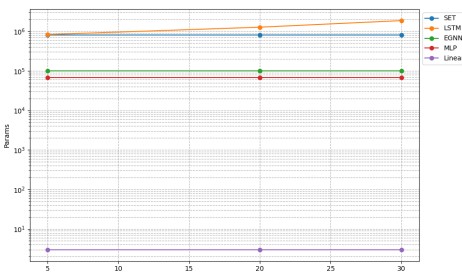

Figure 2: **Left:** Model test MSE versus $N$. **Right:** Number of model parameters versus $N$.

We include in Appendix F results for applying SET to the $N$-body problem with celestial gravitational masses. This demonstrates an example in which the temporal equivariance inductive bias is not appropriate.

## 6 CONCLUSIONS

The imposition of group symmetries on graph neural networks is a promising area of research, demonstrating remarkable real-world results like AlphaFold2 Jumper et al. (2021). However, most research has been centered on spatial equivariance for representational learning on static graphs. For dynamical graph systems, little research has centered on preserving group symmetries across time. We close this gap with the Spacetime $E(n)$-Transformer and show promising results for the $N$-body problem. It will be interesting to see our method applied to harder tasks, such as sequential bio-molecular generation.

Although we chose a graph as the domain of interest, it is plausible to extend notions of spatio-temporal $G$-equivariance to other domains like grids and manifolds. Furthermore, while we leveraged the symmetries of the problem a priori, it may not always be possible to find a simple group for a general problem. Hence, in future work, it would be interesting to learn a group symmetry from underlying data and impose equivariance using methods like LieConv Finzi et al. (2020), which is equivariant to the actions of Lie groups, i.e. the continuous group representation of infinitesimal transformations. Noether's first theorem implies a possible connection to conserved quantities, which was discussed in Alet et al. (2021).

### 6.1 ETHICS

While we only employed simulated datasets, graph neural networks have historically been used in medical and biological applications. In such settings, careful consideration of ethical and responsible data collection is of utmost importance. For instance, in the setting of drug discovery, the way in which we administer synthetically-created drugs on humans and other species must be carefully approached.

### 6.2 REPRODUCIBILITY STATEMENT

All experiments conducted in this paper have been seeded. We selected the best-performing models using Optuna's Bayesian-based hyper-parameter sweep. All relevant hyper-parameters are included in Appendices E and F. We have included a zipped file containing the codebase, which provides details necessary to re-run the results shown in the paper.

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
