# OpenReview forum: "Spacetime $E(n)$-Transformer: Equivariant Attention for Spatio-temporal Graphs"
_ICLR.cc/2025/Conference — ICLR 2025 Conference Withdrawn Submission_

### Official Review · Reviewer_D5RL · 2024-10-26

**Soundness:** 2
**Presentation:** 2
**Contribution:** 1
**Rating:** 3
**Confidence:** 4

**Summary:**

The paper introduces a spatio-temporal graph transformer architecture which incorporates rotation, translation, and permutation equivariance inductive biases for the n-body problem. It assumes the input has a fixed-graph topology, and that the number of nodes in the graphs is fixed, nodes do not appear or dissapear as a function of time. Also, each node is associated with a temporal signal that evolves over time.

**Strengths:**

- The authors combine existing neural network layers into a single architecture for improved performance on the n-body problem.

**Weaknesses:**

- The authors mainly focus on combining existing architectures and layers into a single architecture. They do not provide new theoretical insights for such a combination of layers, nor a purely new processing layer satisfying all required symmetries. This makes the contributions limited.
- There is not enough discussion on positional encodings. I would assume since the authors are using sinusoidal positional encodings that they are referring to positional encodings for the time dimension. However, there is also an extensive set of literature on positional encodings for graph transformers based, for instance, on the Graph Laplacian and other quantities that can be inferred from the graph topology. This are not discussed.
- Also, sinusoidal positional encodings introduce an absolute notion of position which break permutation invariance in the transformer attention operation. The abstract reads: ‘By imposing rotation, translation, and permutation equivariance inductive biases in both space and time’, which probably does not hold in time at least once you add the sinusoidals.
- Regarding Figure 1, the way in which operations are performed is likely suboptimal. Typically positional encodings are added at the very beginning. The Pre-LN transformer formulation is also more widely used than th Post-LN architecture presented in the original attention is all you need paper: meaning, it is more stable to normalize before attention and before the feedforward than after, see: ‘On Layer Normalization in the Transformer Architecture’.
- Experiments are very limited. The architecture seems to be particularly engineered for the n-body problem. There are other problems with similar symmetries but the authors do not carry out additional experiments. Other works in the literature typically use the n-body problem as a synthetic baseline but accompany it with additional experiments, such as for instance top tagging.
- The authors mention they use Optuna’s Bayesian-based hyper-parameter sweep, is this true for all baselines too or only for their model, otherwise results are not directly comparable + reference for Optuna is missing.
- Other previous works that are directly related to the authors line of research and that study the N-body problem are the following: Clifford Group Equivariant Neural Networks, Metric Learning for Clifford Group Equivariant Neural Networks, Neural message passing for quantum chemistry (NMP networks), Tensor Field Networks (which are mentioned in the related work but not added as a baseline). Should be included as baselines.
- The authors could also explore using a hybrid graph transformer framework such as in graph GPS: Recipe for a General, Powerful, Scalable Graph Transformer, which performs local-message passing and global attention in parallel.

**Questions:**

- Not sure how much sense it makes to introduce the message-passing framework in Section 3, equation 1, when the model operates over a fully-connected graph directly. One can view transformers as performing message passing, but typically the equation displayed in the paper is introduced when there is a sense of locality/neighbourhoods induced by an input graph. Since the graph is assumed to be fully connected may be better to use the standard inner-product notation used in Transformers.
- I may be misunderstanding something, but in line 167 the authors mention: ‘which is necessary to model the non-uniform topology of the graphs’, were the authors not assuming a fully-connected graph? In line 112 they mention: ‘we assume a priori that the graph is complete’. Later again in line 236-237: ‘ Since there is no explicit temporal adjacency matrix, we assume a fully connected temporal graph … such a fully connected network does not scale and, instead, we infer the edges of the temporal edges in our mode’. The writing is unclear and the statements are scattered around the text. Could the authors please clarify this and list all their assumptions both in terms of the spatial and temporal connectivity of the graphs in a single paragraph.

---

### Official Review · Reviewer_p4n4 · 2024-10-28

**Soundness:** 3
**Presentation:** 2
**Contribution:** 3
**Rating:** 5
**Confidence:** 3

**Summary:**

This paper introduces a new model called Spacetime E(n)-Transformer (SET), primarily designed to process spatiotemporal graph data. The authors aim to address the limitations of existing spatiotemporal graph neural networks in preserving group symmetries, particularly in temporal and spatial dimensions. The authors utilize EGNN for equivariance and SchNet for invariance.

**Strengths:**

1. The approach is interesting, with the authors designing a novel architecture to achieve equivariance in spatiotemporal graph prediction.

2. The paper is well-organized and easy to follow.

**Weaknesses:**

1. Despite being interesting, the paper's major weakness is the lack of experimental validation. The authors need to demonstrate the method's effectiveness through comparisons with more baseline methods across multiple datasets.

2. More ablation studies are needed to demonstrate the effectiveness of each sub-module, which is currently lacking.

3. While the authors conducted hyperparameter search, reporting model performance under different hyperparameter settings would help better understand the model's capabilities.

In conclusion, I believe this paper requires further improvements to be suitable for publication.

**Questions:**

See above.

---

### Official Review · Reviewer_nCud · 2024-10-31

**Soundness:** 2
**Presentation:** 1
**Contribution:** 2
**Rating:** 3
**Confidence:** 4

**Summary:**

The paper introduces an E(n)-equivariant transformer architecture tailored for spatiotemporal graphs with fixed topologies, specifically complete graphs. The main objective is to embed space-time inductive biases to enhance performance compared to traditional space-time architectures. This model builds on the architecture proposed by Satarras et al., with notable additions such as encoding velocity and performing attention across time. The architecture is tested on the N-body problem dataset, where it demonstrates promising results.

**Strengths:**

The contributions are clearly presented and well-motivated.

**Weaknesses:**

The architecture appears explicitly tailored to the N-body problem, with velocity added as an input feature. A more general description of the architecture, beyond a single application, would strengthen its broader relevance.

Additionally, testing is limited to a single synthetic dataset, which restricts the impact and generalizability of the findings. Evaluating the model on additional datasets, such as those used in Cini et al. (Scalable Spatiotemporal Graph Neural Networks) or in "Long-Range Transformers for Dynamic Spatiotemporal Forecasting,"  by Grigsby et al., would provide a stronger empirical foundation.

The figures could be improved by enlarging the text and providing more detailed captions to aid comprehension.

The related work section and experimental evaluation would benefit from a broader discussion of recent spatiotemporal transformers, including those that may not explicitly include equivariance. Such additions would better contextualize the contribution. Relevant examples include:
Gao, Z., Shi, X., Wang, H., Zhu, Y., Wang, Y. (B.), Li, M., & Yeung, D.-Y. (2022). Earthformer: Exploring space-time transformers for earth system forecasting. Presented at NeurIPS 2022.
Zhao, Z., Chen, Z., Li, J., Xie, X., Chen, K., Wang, X., & Shi, G. STDM-transformer: Space-time dual multi-scale transformer network for skeleton-based action recognition, Neurocomputing.

The paper uses several well known works as building blocks. In this case an ablation study to weight the contribution of each module would make the novel contribution stand out from these works.

**Questions:**

The authors should consider a more rigorous experimental evaluation with recent state-of-the-art spatiotemporal transformer (not necessarily equivariant) and a broader set of datasets.

---

### Official Review · Reviewer_gDvQ · 2024-11-03

**Soundness:** 2
**Presentation:** 1
**Contribution:** 2
**Rating:** 5
**Confidence:** 3

**Summary:**

This paper introduces Spacetime E(n)-Transformer (SET), a E(n)-equivariant  spatio-temporal model suited to model spatio-temporal graphs. SET first updates the feature, position and velocity of each node at time t using E(n)-equivariant spatial attention by combining a SchNet continuous filter and a EGNN layer.  To account for temporal information,  the paper introduces the Equivariant Temporal Attention Layer (ETAL), which consists in a self-attention layer across time for node features and velocities, and a temporal neighborhood attention for positions. Finally, instead of having a fixed adjacency matrix, SET uses a temporal self-attention on adjacency matrices. SET is evaluated on the (charged) N-body problem and shows improvement over the selected baselines.

**Strengths:**

**Novelty**: The paper effectively extends E(n)-equivariance to temporal data through the Equivariant Temporal Attention Layer (ETAL) which is novel. The idea of leveraging self-attention mechanism across time is natural and very simple (which is positive).

**Theoretical Consistency**: The model is theoretically sound in maintaining E(n)-equivariance across both space and time.

**Weaknesses:**

**Quality of Writing**: The paper’s organization is somewhat unclear, with overlapping notations that may make it challenging for readers to follow. Additionally, the main contribution is not clearly highlighted in the introduction and the method section, which could improve the paper’s overall clarity.

**Experimental Evaluation**: The experimental setup might benefit from stronger baseline comparisons, more comprehensive ablations, and clearer explanations regarding certain model components. Further rigor in these areas would help substantiate the proposed method.

Please the questions below for more detailed comments / questions.

**Questions:**

- Since the E(n)-equivariant spatial attention section primarily builds on existing work, it might be more suitable in the background section rather than being positioned as a main contribution.

- Effectiveness of Temporal Adjacency: In the charged N-body experiment, the temporal adjacency component does not seem to contribute as expected. The explanation that “edge attributes contain information about charges” suggests a possible shortcut in your model, potentially making it difficult to learn the self-attention projections. Testing with identity matrices for the query, key, and value projections could help clarify this observation.

- Ablation Study Clarifications: In the ablation table, the meaning of “Equiv=True/False” could be clarified, as it is currently difficult to interpret how equivariance is deactivated. Additionally, the rationale behind combining SchNet[6] with EGNN [5] could be explained further. Could you elaborate on this particular choice of spatial encoders. An ablation on this aspect could be interesting.

- Baselines and Comparisons: The main results (Table 2) are based on relatively simple baselines, such as LSTM without spatial encoding, EGNN without temporal encoding, MLP, and Linear (which are quite trivial baselines). Including a stronger baseline, perhaps by replacing the ETAL layer with an LSTM, could better highlight the impact of E(n)-equivariance in temporal encoding. Furthermore, why not comparing against SE(3)-Transformer[4], EGNN + SchNet, LieConv [3], LNNs [2], DynGCN [1], DynGFormer [7] and ESTAG [8]?

Other comments / questions:
- In Eq 1,  phi and psy are not described. If this equation is not part of your contribution, why not place it in the background section?
- In line 133, shouldn't it be attention instead of self-attention? because it is between node i and its neighbors j.
- In eq 6 shouldn't it be $h_i^{(l)}$ instead of $h_i^{(l+1)}$ in the SchNet ? You may consider presenting the general architecture of SET at the beginning of the method section to help the reader understand each component better.
- In line 209, is it E(n)-invariant or E(n)-equivariant ?
- Can you elaborate on why the projection matrices are fixed to the identity for the velocity self-attention ?
- While I understand the purpose of $\phi_{inf}$, it seems it is not describe in the paper. Is it the same as in EGNN [5] ?


References:

[1] Choi et al, "Dynamic graph convolutional networks with attention mechanism for rumor detection on social media"

[2] Cranmer et al; "Lagrangian neural networks"

[3] Finzi et al, "Generalizing convolutional neural networks for equivariance to lie groups on arbitrary continuous data"

[4] Fuchs et al,"Se(3)-transformers: 3d rototranslation equivariant attention networks"

[5] Satorras et al, "E(n) equivariant graph neural networks"

[6] Schütt et al, "Schnet – a deep learning architecture for molecules and materials"

[7] Yu et al, "Towards better dynamic graph learning: New architecture and unified library"

[8] Wu et al, "Equivariant spatio-temporal attentive graph networks to simulate physical dynamics"

---

### Official Review · Reviewer_teMR · 2024-11-04

**Soundness:** 3
**Presentation:** 2
**Contribution:** 2
**Rating:** 3
**Confidence:** 3

**Summary:**

The paper introduces an architecture for modelling spatio-temporal graphs, which is equivariant to translation and rotation of the nodes. It relies on applying Equivariant Graph Convolutional Network on the spatial dimension (independent for each timestep) followed by an equivariant transformer on the temporal dimension (independent for each node). The model is compared against a series of baselines on a synthetic N-body system dataset.

**Strengths:**

- Studying equivariant architectures for spatio-temporal graphs is an interesting and novel direction.

**Weaknesses:**

- My main concern regards the limited novelty of the proposed model. Architectures for spatio-temporal processing that use a factorization approach (separating spatial and temporal processing) are already established (e.g., DynGCN). While the paper claims to be the first to introduce equivariance in this context, it independently applies equivariance in both the spatial and temporal dimensions using an existing method (Santorras et al.). No specific modifications to the model are required; simply applying this approach first to the spatial dimension,
 and than to the fully connected temporal graph results in an E(n)-equivariant model. Consequently, the contribution is mainly incremental, arising from a straightforward combination of existing methods, which I find insufficient to recommend for publication.
- The main focus of the paper is to introduce the first equivariant architecture tailor for spatio-temporal graphs. However,  it is unclear what unique challenges the temporal dimension introduces. From the proposal, both the temporal and the spatial dimenion are processed in a mirrored way, suggesting that the temporal dimension may not pose significant additional challenges when working with geometric graphs.
- The experimental section does not compare against any existing spatio-temporal methods in the literature (such as the one mentioned in the related work section). Is there any particular reason why that is not possible? The only comparison are against very weak baselines that either ignore interactions entirely (LSTM) or ignore temporality entirely (e.g. spatial-level EGNN).
- The manuscript can be improved in terms of clarity. For example, abbreviation such as SATT and TATT used in the ablation section are not explained. The captions of the figures are braodly non-informative (e.g. table 2). The results presented in the experimental sections are not thoroughly discussed. Minor: citations does not follow the ICLR guideline (regarding when to use citep vs citet). In some places this makes the reading hard to parse.
- It would be beneficial to include additional datasets, ideally non-synthetic ones. However, using only the n-body dataset would be acceptable if comparisons with existing models are conducted in a highly thorough and rigorous manner.

**Questions:**

Please see the sections above.

---

### Official Review · Reviewer_VAKn · 2024-11-06

**Soundness:** 2
**Presentation:** 1
**Contribution:** 1
**Rating:** 3
**Confidence:** 4

**Summary:**

The paper proposes an E(n)-equivariant Transformer architecture for spatio-temporal data, called Spacetime E(n)-Transformer (SET). The proposal builds upon E(n) Equivariant GNNs and temporal attention mechanisms. Experiments on the N-body dataset show the effectiveness of the proposed model compared to MLPs, EGNNs, and LSTMs.

**Strengths:**

**Relevance**: The problem of modeling the dynamic of physical systems using neural nets is a very relevant research topic.

**Ablation study**: The paper conducts an ablation study that shows the impact of different model components. The results confirm the importance of the full model and equivariance.

**Weaknesses:**

**Exposition/Organization**: Overall, the paper could better use the available space (page limit). For instance, Section 2.2 could be significantly compressed. Also, I would move Algorithms 1 and 2 to Appendix --- after strengthening the description of the method section, these algorithms are unnecessary. The same applies to the squared loss in Eq. (14). The description of EGNNs (EGCL layers) should be in the Background. Similarly, several basic definitions/notations are provided in the method section when they should be in the Background.

**Novelty**: Overall, I found the novelty of this paper very limited. From a methodological perspective, the proposal consists of a combination of Equivariant GNNs (SchNet) and temporal models that preserve equivariance. Being ETAL the main methodological contribution, I believe the paper should extensively evaluate it in different settings. From a conceptual level, this is not the first work to consider equivariant spatial and equivariant temporal models to simulate dynamical physical systems.

**Baselines/experiments**: One of the main weaknesses is the experimental setup. The paper only considers a single task (the N-body problem), which may cause readers to interpret it as limited applicability. Also, the paper does not consider any model for spatiotemporal data (e.g., with vanilla GNNs replaced with equivariant ones) as a baseline in the experiments. The method in [1] is also not considered in the comparison.

[1] Wu et al., Equivariant Spatio-Temporal Attentive Graph Networks to Simulate Physical Dynamics, NeurIPS, 2023.

**Questions:**

1. It seems that ETAL is one of the main contributions of the paper. Can we combine ETAL (without the geometric features) with vanilla GNNs for standard (non-geometric) spatiotemporal benchmarks?

2. Could the authors elaborate on the motivation for sequentially applying SchNet and EGCL? Also, in Eq. (6), $h_i^{(l+1)}(t)$ appears as the input and output of ScnNet. A similar issue occurs in Algorithm 1 (Line 283), where $h_i^{(l+1)}(t)$ gets overwritten.

3. How does the proposed approach compare with the method in [1]? Also, how does the proposal perform on the molecular and protein datasets in [1]?

[1] Wu et al., Equivariant Spatio-Temporal Attentive Graph Networks to Simulate Physical Dynamics, NeurIPS, 2023.

---

### Note · Authors · 2024-11-15

**Comment:**

I am withdrawing the paper as I do not believe it currently meets the standards of the conference.

**Withdrawal Confirmation:**

I have read and agree with the venue's withdrawal policy on behalf of myself and my co-authors.